

# Efficacy and safety of low-molecular-weight-heparin plus citrate in nephrotic syndrome during continuous kidney replacement therapy: retrospective study

Di Wang[1,2,3,4,5,6], Mengqin Yang[1,2,3,4,5,6], Siyuan Li[1,2,3,4,5,6], Can Tang[1,2,3,4,5,6], Jun Ai[1,2,3,4,5,6] and Diankun Liu[1,2,3,4,5,6]

[1] Department of Nephrology, Nanfang Hospital, Southern Medical University, Guangzhou, Guangdong, China
[2] National Clinical Research Center for Kidney Disease, Nanfang Hospital, Guangzhou, Guangdong, China
[3] State Key Laboratory of Organ Failure Research, Southern Medical University, Guangzhou, Guangdong, China
[4] Guangdong Provincial Institute of Nephrology, Guangzhou, Guangdong, China
[5] Guangdong Provincial Key Laboratory of Renal Failure Research, Guangzhou, Guangdong, China
[6] Guangzhou Regenerative Medicine and Health Guangdong Laboratory, Guangzhou, Guangdong, China

Corresponding authors
Jun Ai, aij1980@163.com
Diankun Liu, tennisldk@163.com

## ABSTRACT

**Background:** Nephrotic syndrome (NS) is a condition often necessitating continuous kidney replacement therapy (CKRT) due to severe edema and other complications. Anticoagulation is critical in CKRT to prevent filter clotting, with regional citrate anticoagulation (RCA) being the first-line method. However, the hypercoagulable state of NS may require alternative strategies. Optimal anticoagulation therapy in NS patients undergoing CKRT is lacking.

**Methods:** This retrospective observational study included 251 CKRT sessions from 122 NS patients treated at Nanfang Hospital, Southern Medical University, from January 2019 to December 2022. Patients were divided into three groups based on anticoagulation method: Low-molecular-weight-heparin (LMWH) alone, RCA alone, and RCA plus LMWH. Filter lifespan, incidence of filter clotting, and adverse events were assessed to evaluate the efficacy and safety profiles of each anticoagulation methods.

**Results:** The combination of RCA and LMWH demonstrated a significantly longer mean filter lifespan and lower incidence of filter clotting compared to LMWH or RCA alone. RCA plus LMWH also showed a lower incidence of overall adverse events, particularly thrombosis, without an increase in bleeding complications. Multivariate Cox analysis indicated that RCA plus LMWH was particularly effective in patients with normal kidney function.

**Conclusions:** RCA combined with LMWH provides a superior anticoagulation strategy in NS patients undergoing CKRT, with enhanced filter lifespan and reduced clotting and thrombotic events without increasing bleeding risk. Further research is needed to optimize dosing and validate these results in broader populations.

## INTRODUCTION

Nephrotic syndrome (NS) is a clinical condition characterized by significant proteinuria and hypoalbuminemia, often accompanied by peripheral edema, hypertension, and kidney dysfunction (*Wada et al., 2021*; *Politano, Colbert & Syndrome, 2020*). In severe cases, where edema is resistant to diuretics, kidney replacement therapy may be required. Continuous kidney replacement therapy (CKRT) is preferred due to its ability to facilitate gradual ultrafiltration (*Bouchard & Mehta, 2009*; *Brandenburger et al., 2017*). Anticoagulation is essential during CKRT to prevent clotting within the filter and extracorporeal circuit. Unfractionated Heparin (UFH) and low-molecular-weight-heparin (LMWH) are the most commonly used anticoagulants; however, their use is complicated by the hypercoagulable state of NS patients, necessitating higher doses and increasing bleeding risks (*MacEwen, Watkinson & Winearls, 2015*; *Kerlin, Ayoob & Smoyer, 2012*; *International Society of Nephrology, 2012*).

Regional citrate anticoagulation (RCA), achieved by chelating calcium and thus inhibiting the coagulation cascade, has emerged as a viable alternative, with studies confirming its safety and efficacy in CKRT (*International Society of Nephrology, 2012*; *Morgera et al., 2009*; *Zhang & Hongying, 2012*; *Liu et al., 2016*; *Mehta, 1994*; *Kindgen-Milles, Brandenburger & Dimski, 2018*). Despite its benefits in maintaining extracorporeal anticoagulation, citrate does not address the heightened risk of venous thrombosis due to prolonged bed rest and catheter placement in NS patients undergoing CKRT.

Recent research suggests that combining heparin with RCA may extend filter lifespan in critically ill patients (*Hetzel et al., 2011*; *Wu et al., 2015*; *Shankaranarayanan et al., 2020*). However, studies focusing on NS patients are lacking. This retrospective cohort study aims to evaluate the efficacy and safety of combining LMWH with RCA in NS patients undergoing CKRT, seeking to identify a coagulation therapy that minimizes venous thrombosis and maximizes filter lifespan.

## MATERIALS AND METHODS

### Study population and clinical characteristics

We included 178 patients (aged over 14 years) diagnosed with NS, undergoing 373 CKRT sessions, at the Department of Nephrology, Nanfang Hospital, Southern Medical University, from January 2019 to December 2022. A total of 50 sessions without anticoagulant therapy, 54 sessions with incomplete baseline or dialysis-related data and 18 sessions with oral anticoagulants were excluded (shown in Fig. S1). Consequently, 251 CKRT sessions of 122 patients were enrolled in the final analysis. Clinical and biological data, including age, sex, comorbidities, weight, height, 24 h urine total protein (24 h UTP), serum albumin (ALB), blood urea nitrogen (BUN), serum creatinine (SCr) with estimated glomerular filtration rate (eGFR) estimated using Chronic Kidney Disease Epidemiology Collaboration (CKD-EPI) equation, coagulation test including prothrombin time (PT),

prothrombin time-international normalized ratio (PT-INR), activated partial thromboplastin time (APTT) and D-Dimer, serum calcium ($Ca^{2+}$), hemoglobin (HGB), platelet count (PLT) and comorbidities, were collected from medical records and laboratory databases, respectively. Patients were categorized into three groups based on their anticoagulation therapy during CKRT: LMWH group (Group A), RCA group (Group B), and a combination of LMWH and RCA (Group C).

The study protocol adhered to the Declaration of Helsinki, received approval and ethical exemption from obtaining informed consent from patients due to the retrospective nature of the study from the medical ethics committee of Nanfang Hospital, Southern Medical University. The approval number is NFEC-2019-213.

## CKRT treatment

CKRT was administered using pump-driven devices (Prismaflex-Gambro Lundia, Lund, Sweden) with fluid balance system and a biocompatible membrane measuring 1.0 m$^2$ (M-100; Gambro Lundia, Lund, Sweden). Modes of CKRT included slow continuous ultrafiltration (SCUF), continuous venovenous haemodialysis (CVVHD), continuous venovenous haemofiltration (CVVH), and continuous venovenous haemodiafiltration (CVVHDF). A standard dose of 2,000 ml/h was delivered to all the patients except for those on SCUF therapy. The blood flow rate ranged from 100 to 200 ml/min. The ultrafiltration rate was adjusted by nephrologists based on clinical criteria, with a maximum rate of 500 ml/h. The filter was replaced after 24 h in line with the manufacturer's guidelines in the absence of clotting. Subsequently, a double-lumen catheter with 11.5F in diameter and a length varying from 16 to 20 cm (ABLE; Guangdong Baihe Medical Technology Co., Ltd, Nanhai District, Foshan City, China), was inserted *via* a central vein, choosing either the jugular or the femoral vein for access.

A commercially prepared hemofiltration basic solution (Chengdu Qingshan Likang Pharmaceutical Co., Ltd, Chengdu, China) served as the dialysate and replacement fluid. In Group A, 4,000 ml of this solution combined with 250 ml of 5% $NaHCO_3$ fluid was applied to achieve a balanced, potassium-free solution. The resulting solution contained 141 mmol/L sodium, 1.5 mmol/L calcium, 35 mmol/L $HCO_3^-$, 0.75 mmol/L magnesium, 110 mmol/L chloride, and 10 mmol/L glucose. In the remaining two groups, $NaHCO_3$ was administered *via* a distinct intravenous route at a reduced dose, attributable to the *in vivo* conversion of citrate into bicarbonate. Potassium chloride (KCl) was added if the patient has a risk of hypokalemia.

Anticoagulation therapy was tailored by clinical physicians to each patient's individual characteristics and therapeutic objectives, provided no contraindications were present. LMWH was administered into the circuit before the filter initially, and re-infused every 4–8 h to maintain the patency of the circuit, with the total dosage varying between 0.1 and 2 ml per circuit. Concurrently, a pre-filter infusion of a 4.0% sodium citrate solution (Chengdu Qingshan Likang Pharmaceutical Co., Ltd, Chengdu, China) at an initial flow rate of 160–200 ml/h provided anticoagulation in extracorporeal circuit. This rate might subsequently be adjusted in response to the measured levels of post-filter ionized calcium if available.

We meticulously documented the lifespan of the filter, which we defined as the interval from initiation to a non-elective cessation of the circuit, necessitated by filter clotting or reaching the conclusion of treatment in the absence of clotting. Filter clotting was specifically defined as instances where the filter lifespan was curtailed relative to the projected duration of therapy, attributable to clot formation.

## The definition of adverse effects of the catheter

Catheter related infection: With the clinical manifestation of infection symptoms, confirmed either by the presence of bacteria in cultures from the catheter tip and/or peripheral blood, or by the exclusion of other underlying causes in cases where microbial cultures are negative (*Ren et al., 2019*).

Catheter dysfunction: Manifested as either catheter prolapse or kinking, which leads to a reduction in blood flow or renders the catheter unusable until it is replaced (*Moist, 2016*).

Thrombosis: Diagnosed through ultrasonography, thrombosis is identified by the formation of a fibrin sleeve encasing the catheter or a thrombus adhering to the vessel wall or in the catheter. Inadequate catheter blood flow requiring the application of a urokinase seal, with or without the clinical signs of deep venous thrombosis, was also included (*Moist, 2016*; *Wang et al., 2016*).

Bleeding: Blood exudation around the site of the dialysis catheter insertion (*Lazrak et al., 2017*).

## Statistical analysis

All collected data for this study were meticulously cataloged within a standardized Excel database. The statistical analysis was conducted utilizing SPSS software, version 26.0 (IBM Inc., Armonk, NY, USA), GraphPad Prism, version 8.0.2.263 (GraphPad Software, San Diego, CA, USA), and R Studio, version 4.1.2. Continuous variables were articulated as mean (standard deviation, SD) or median (inter-quartile range, IQR), while categorical variables were delineated as frequencies and percentages (n (%)). Group comparisons for continuous data were executed *via* one-way ANOVA or rank sum test if not conform to normal distribution, whereas categorical data were assessed using the $\chi^2$ test or Fisher's exact test, contingent upon suitability. Kaplan-Meier analysis was employed to construct survival curves for filter lifespan across different groups, with the log-rank test elucidating disparities. Binary logistic regression analysis was applied to determine the correlation between anticoagulation methods and circuit clotting. Both univariate and multivariate cox proportional hazards analysis were used to explore the risk factors associated with circuit clotting. Odds ratios (ORs), hazard ratios (HRs) and the corresponding 95% confidence intervals (CIs) were calculated separately, with *P* value of less than 0.05 denoting statistical significance.

## RESULTS

### Patients

In this study, we meticulously screened a total of 373 CKRT sessions from 178 NS patients treated at Nanfang Hospital, Southern Medical University, from January 2019 to
**Table 1 Baseline characteristics of study population.**

| | Group A (LMWH) | Group B (RCA) | Group C (LMWH + RCA) | P value |
|---|---|---|---|---|
| Numbers of patients | 84 | 10 | 28 | |
| Demographics | | | | |
| Age, median (IQR), y | 47.7 (24.0–65.0) | 45.8 (23.0–67.0) | 42.3 (21.0–56.0) | 0.50 |
| Male, No. (%) | 63 (75.0) | 4 (40.0) | 20 (71.4) | 0.07 |
| Height, mean (SD), cm | 164.5 (8.3) | 160.5 (8.4) | 164.5 (8.1) | 0.43 |
| Weight, median (IQR), kg | 70.9 (61.9–79.2) | 68.8 (63.8–80.6) | 74.5 (66.1–85.0) | 0.40 |
| BSA[a], mean (SD), $m^2$ | 2.4 (0.2) | 2.3 (0.2) | 2.4 (0.2) | 0.43 |
| BMI, mean (SD), $kg/m^2$ | 26.2 (4.3) | 26.8 (4.5) | 27.6 (4.4) | 0.39 |
| Laboratory data | | | | |
| 24 h UTP, median (IQR), g/24 h | 12.9 (6.5–18.7) | / | 14.7 (7.5–22.1) | 0.55 |
| ALB, median (IQR), g/L | 20.8 (16.8–27.6) | 19.7 (16.5–25.6) | 19.9 (19.0–26.9) | 0.89 |
| SCr, median (IQR), μmol/L | 300.1 (140.8–393.5) | 211.8 (135.5–346.3) | 292.2 (213.0–390.0) | 0.40 |
| eGFR[b], median (IQR), ml/min · 1.73 $m^2$ | 34.2 (12.3–42.8) | 38.4 (15.2–51.6) | 34.1 (11.6–31.8) | 0.91 |
| BUN, median (IQR), μmol/L | 18.6 (11.2–25.9) | 17.4 (10.2–22.6) | 21.3 (18.9–33.4) | 0.42 |
| HGB, median (IQR), g/L | 106.4 (85.5–131.5) | 88.0 (77.5–127.0) | 116.0 (93.5–131.5) | 0.15 |
| PLT, mean (SD), $\times 10^9$/L | 279.3 (99.0) | 168.3 (154.3) | 268.6 (120.5) | <0.05 |
| PT, median (IQR), s | 11.0 (10.1–11.3) | 12.0 (10.2–11.6) | 11.3 (9.7–11.5) | 0.28 |
| PT-INR, median (IQR) | 1.0 (0.9–1.0) | 1.1 (0.9–1.0) | 1.0 (0.8–0.9) | 0.23 |
| APTT, median (IQR), s | 28.9 (23.2–32.5) | 31.3 (24.0–33.9) | 29.7 (23.4–32.9) | 0.66 |
| D-Dimer, median (IQR), μg/ml FEU | 5.3 (1.6–7.7) | 5.5 (2.4–7.8) | 6.0 (2.6–4.4) | 0.86 |
| $Ca^{2+}$, median (IQR), mmol/L | 1.9 (1.8–2.0) | 2.0 (1.8–2.0) | 1.9 (1.8–2.1) | 0.48 |
| Comorbid diseases | | | | |
| HTN, No. (%) | 47 (56.0) | 8 (80.0) | 17 (60.7) | 0.336 |
| DM, No. (%) | 24 (28.6) | 3 (30.0) | 14 (50.0) | 0.112 |
| CVDs, No. (%) | 10 (11.9) | 0 | 1 (3.6) | 0.622 |
| HBV, No. (%) | 7 (8.3) | 1 (10.0) | 3 (10.7) | 0.924 |
| Malignant tumor, No. (%) | 4 (4.8) | 0 | 0 | 0.756 |

**Notes:**
The above results were calculated based on the population of each group.
Abbreviations: LMWH, Low molecular weight heparin; BSA, body surface area; BMI, body mass index; 24 h UTP, 24 h urinary total protein; ALB, albumin; SCr, serum creatinine; BUN, blood urea nitrogen; HGB, hemoglobin; PLT, platelet; PT, prothrombin time; PT-INR, prothrombin time-international normalized ratio; APTT, activated partial thrombin time; $Ca^{2+}$, serum calcium levels; HTN, Hypertension; DM, Diabetes mellitus; CVDs, cardiovascular disease and/or cerebrovascular disease; HBV, hepatitis B viral infection.
[a] BSA is according to the general formula of Chinese human body surface area, the formula is *0.010061 \* height ($m^2$) + 0.010124 \* weight (kg) − 0.010099.*
[b] eGFR is calculated according to the 2009 CKD-EPI equations.

December 2022. Exclusions were made for 50 sessions that did not involve anticoagulant therapy, 54 sessions with incomplete baseline or dialysis-related data and 18 sessions with oral anticoagulants. Consequently, 251 CKRT sessions from 122 patients were included in the analysis. A total of 87 males and 35 females among them, median age was 52.0 years, the average of body mass index (BMI) was 26.6 ± 0.4 $kg/m^2$. Patients were categorized based on the anticoagulation method employed: Group A consisted of 84 patients with 185 circuits using LMWH, Group B comprised 10 patients with 31 circuits using RCA, and Group C included 28 patients with 35 circuits using a combination of RCA and LMWH.

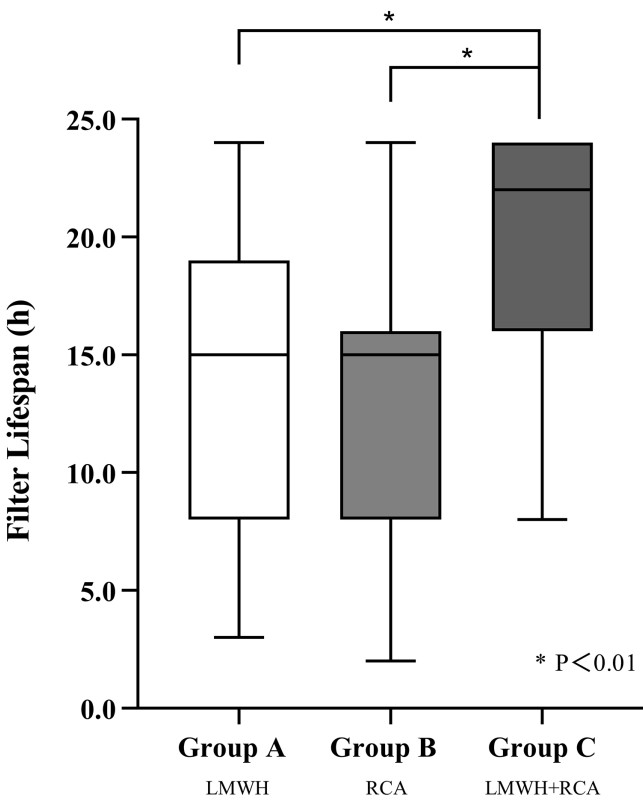

**Figure 1 Filter lifespan among three groups.** Abbreviations: LMWH, Low molecular weight heparin; RCA, regional citrate anticoagulation. *P < 0.01.     

Table 1 presents the baseline features of the study population. The comparative analysis revealed no statistically significant differences among the three groups in terms of age, sex, height, weight, BMI, body surface area (BSA) and a spectrum of laboratory markers, including 24 h UTP, ALB, SCr, eGFR, BUN, HGB, coagulation test parameters and serum $Ca^{2+}$ levels. Within Group A, there were 31 patients aged over 60 years, 51 patients with eGFR $\leq$ 30 ml/min $\cdot$ 1.73 $m^2$, and 46 patients with ALB levels below 20 g/L. Correspondingly, in Groups B and C were 3, 6, and 7 patients, and 7, 16, and 16 patients, respectively (Table S1).

### Primary outcomes

Group C demonstrated a significantly longer median filter lifespan (22 h) compared to Group A (16 h) and B (15 h) ($P < 0.01$), group C also showed a higher cumulative filter survival rate ($P < 0.01$) (shown in Figs. 1 and 2). The incidence of filter clotting was lower in Groups B and C compared to Group A, with respective rates of 28.1%, 9.7%, and 5.7% ($P < 0.01$). No significant difference was found between Groups B and C ($P = 0.659$), (shown in Table 2).

### Secondary outcomes

All the patients received 2,000 ml/h CKRT dose except for those on SCUF therapy. The median delivered ultrafiltration rates varied across the groups, with Group C achieving

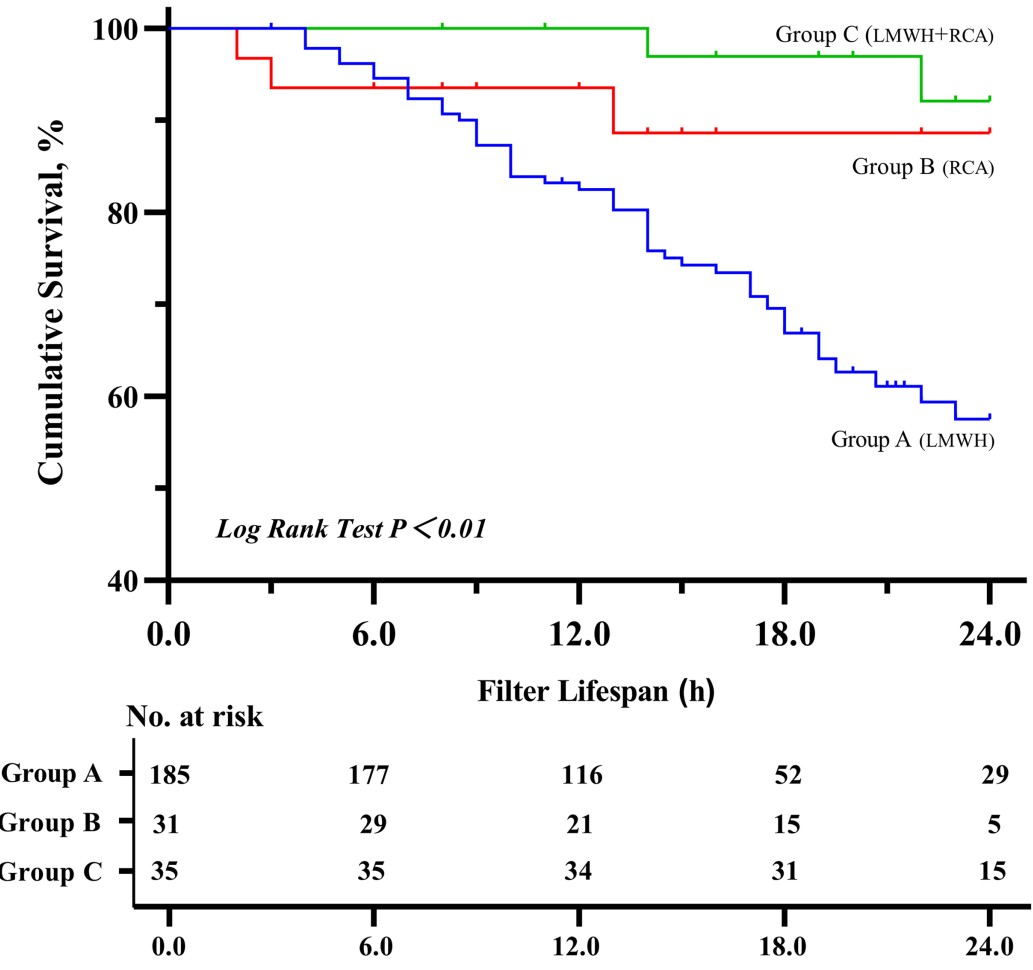

**Figure 2 Kaplan–Meier estimated probability of circuit clotting-free survival on the basis of different anticoagulation methods.** Abbreviations: LMWH, Low-molecular-weight-heparin; RCA, regional citrate anticoagulation.

higher ultrafiltration (333.7 ml/h). Group C also used a significantly lower daily dosage of LMWH compared to Group A, and a lower initial citrate infusion rate compared to Group B (shown in Table 2).

## Incidence of adverse events

Adverse events were assessed per CKRT session (shown in Table 3). Group A experienced the highest number of circuits with adverse events (50), followed by Group B (23) and C (four), correlating to 27.0%, 74.2%, and 11.4% of their total circuits, respectively ($P < 0.01$). Notably, multiple adverse events occurred within 23 sessions. In Group A, thrombosis, catheter dysfunction, and bleeding were the predominant catheter-related adverse events, occurring in 22, 20, and eight circuits, with incidences of 11.9%, 10.8%, and 4.3%, respectively. Dialysis-related adverse events such as dizziness/hypotension, myalgias/spasms, and nausea/vomiting were also noted, with incidences of 4.3%, 2.2%, and 2.2%, respectively. The most frequent catheter-related adverse event of Group B was thrombosis, affecting 17 circuits (54.8%), along with two instances of catheter dysfunction and one of

**Table 2 Efficacy of different anticoagulation therapy.**

| | Group A (LMWH, 185) | Group B (RCA, 31) | Group C (LMWH + RCA, 35) | P value |
|---|---|---|---|---|
| Primary outcomes | | | | |
| Filter clotting, No. (%) | 52 (28.1) | 3 (9.7) | 2 (5.7) | <0.01 |
| Filter lifespan[a], median (IQR), h | 16.0 (8.0–20.3) | 15.0 (8.0–16.0) | 22.0 (16.0–24.0) | <0.01 |
| Secondary outcomes | | | | |
| Delivered CKRT dose, median (IQR), ml/h·kg | 29.4 (26.4–32.3) | 32.3 (26.5–32.3) | 27.9 (21.8–29.9) | <0.01 |
| Initial infusion rate of Citrate, median (IQR) ml/h | / | 180.0 (180.0–180.0) | 160.0 (160.0–160.0) | <0.01 |
| Daily dose of LMWH[b], median (IQR), AXaIU | 12,000 (10,143–16,650) | / | 9,000 (8,000–12,000) | <0.01 |
| Ultrafiltration, median (IQR), ml/h | 323.4 (255.0–376.2) | 211.7 (129.1–334.1) | 333.7 (281.4–395.3) | <0.01 |

Notes:
Abbreviations: LMWH, Low molecular weight heparin; CKRT, continuous kidney replacement therapy.
[a] Filter lifespan was defined as the interval from initiation to a non-elective cessation of the circuit, necessitated by filter clotting or reaching the conclusion of treatment in the absence of clotting
[b] Daily dose was calculated by averaging the cumulative LMWH dose of each circuit by the actual treatment time of each cycle, and then multiplied it by 24 h.

**Table 3 Adverse events of different anticoagulation therapy.**

| | Group A (LMWH, 185) | Group B (RCA, 31) | Group C (LMWH + RCA, 35) | P value |
|---|---|---|---|---|
| Circuit with adverse event[a], No. (%) | 50 (27.0) | 23 (74.2) | 4 (11.4) | <0.01 |
| Overall Adverse events, No. | 73 | 25 | 6 | |
| Catheter adverse events, No. (%) | 52 (71.2) | 20 (80.0) | 3 (50.0) | |
| Catheter associated infections | 2 (1.1) | 0 | 0 | |
| Thrombosis | 22 (11.9) | 17 (54.8) | 1 (2.9) | <0.01 |
| Bleeding | 8 (4.3) | 1 (3.2) | 0 | 0.45 |
| Catheter dysfunction | 20 (10.8) | 2 (6.5) | 2 (5.7) | 0.53 |
| Dialysis adverse events, No. (%) | 21 (28.8) | 5 (20.0) | 3 (50.0) | |
| Dysphoria | 1 (0.5) | 0 | 0 | |
| Weakness | 1 (0.5) | 0 | 0 | |
| Dizziness/Hypotension | 8 (4.3) | 1 (3.2) | 2 (5.7) | 0.88 |
| Perioral numbness | 0 | 0 | 0 | |
| Palpitation/Chest pain | 0 | 1 (3.2) | 0 | |
| Hypothermia/Shiver | 1 (0.5) | 1 (3.2) | 0 | 0.25 |
| Dry cough | 1 (0.5) | 0 | 0 | |
| Nausea/Vomiting | 4 (2.2) | 1 (3.2) | 0 | 0.61 |
| Stomachache | 1 (0.5) | 0 | 0 | |
| Myalgia/Spasm | 4 (2.2) | 1 (3.2) | 1 (2.9) | 0.92 |

Notes:
[a] Circuits with adverse events were defined as that CKRT circuit in which one or more adverse events were observed. Irrespective of the quantity of adverse events documented within a given circuit, each circuit is counted singularly as one with adverse events.
Abbreviations: LMWH, Low molecular weight heparin.

bleeding. In Group C, a lower incidence of both catheter and dialysis-related adverse events was observed: one thrombosis, two catheter dysfunctions, two case of dizziness/hypotension, and one of myalgia/spasm. The rate of thrombosis in Group C was significantly less than that in Groups A and B ($P < 0.01$) with no bleeding events recorded.

**Table 4 The correlation between anticoagulation therapy and filter clotting in univariate and multivariate logistic analysis.**

| Subgroup | Total circuits | Circuit clotting | Unadjusted OR (95% CI) | P value | Model 1[a] | | | | Model 2[b] | | | | Model 3[c] | | | |
|---|---|---|---|---|---|---|---|---|---|---|---|---|---|---|---|---|
| | | | | | Adjusted OR (95% CI) | P value | AIC | R[2] | Adjusted OR (95% CI) | P value | AIC | R[2] | Adjusted OR (95% CI) | P value | AIC | R[2] |
| Group A (LMWH) | 185 | 52 | Reference | 0.012 | Reference | 0.009 | 182.06 | 0.159 | Reference | 0.023 | 171.92 | 0.315 | Reference | 0.029 | 179.53 | 0.329 |
| Group C (LMWH + RCA) | 35 | 2 | 0.155 [0.036–0.669] | | 0.125 [0.027–0.589] | | | | 0.158 [0.032–0.776] | | | | 0.161 [0.031–0.828] | | | |

**Notes:**
Abbreviations: LMWH, Low molecular weight heparin; RCA, regional citrate anticoagulation; OR, odd ratio.
[a] Model 1: adjusted for age, gender, baseline BMI.
[b] Model 2: adjusted for covariates in model 1 and serum albumin, serum creatinine, eGFR (CKD-EPI), blood urea nitrogen, hemoglobin, platelet and serum calcium levels.
[c] Model 3: adjusted for covariates in model 2 and catheter site, catheter length, prothrombin time, activated partial thrombin time and D-Dimer.

Furthermore, no severe adverse events, such as deaths or ICU admissions, were reported across all groups.

### Correlation between anticoagulation and filter clotting

Univariate and multivariate logistic analysis indicated that RCA + LMWH anticoagulation was superior to LMWH to prevent filter clotting. The same results were shown in certain patient demographics and clinical profiles, including male patients, those under 60 years of age, with BMI over 24.0 kg/m$^2$, eGFR above 30 ml/min · 1.73 m$^2$, and HGB over 90 g/L (shown in Table 4 and Table S1).

### The risk factors for filter clotting

Table 5 outlined the risk factors for filter clotting using univariate and multivariate Cox analysis. Univariate analysis identified low levels of serum ALB and BUN, high levels of eGFR, HGB and PLT as contributors to filter clotting. The multivariate analysis using different models conclusively demonstrated that LMWH combined with RCA contribute to the lowest incidence of filter clotting compared to LMWH or RCA alone. Upon expanding the range of inclusion factors, only an elevated eGFR (HR 1.03, 95% CI [1.01–1.05], $P < 0.01$) remained a significant predictor of filter clotting.

## DISCUSSION

RCA is established as the primary anticoagulation strategy in CKRT (*Brandenburger et al., 2017*; *Ostermann et al., 2020*). However, its efficacy in the NS population still need to be explored. Given the inherent hypercoagulable state associated with NS (*Mirrakhimov et al., 2014*), RCA may not be the optimal anticoagulation modality. To the best of our knowledge, this retrospective observational study is the first to compare the efficacy of RCA in conjunction with LMWH against traditional anticoagulation therapies in NS patients. Our findings indicate that the combination of RCA and LMWH is effective in patients undergoing CKRT, as evidenced by an extended filter lifespan, a reduced incidence of filter clotting, and enhanced safety profiles, including a lower incidence of thrombosis and bleeding events.

NS is a prevalent clinical syndrome, accounting for approximately 40% of kidney biopsy cases (*Sugiyama et al., 2011*; *Zhou et al., 2011*). Severe, diuretic-resistant edema, a common

**Table 5 Risk factors for filter clotting analysis in nephrotic syndrome patients.**

| Factor | Univariate analysis | | Multivariate analysis | | | | | |
| --- | --- | --- | --- | --- | --- | --- | --- | --- |
| | | | Model 1[a] | | Model 2[b] | | Model 3[c] | |
| | HR (95% CI) | P value | HR (95% CI) | P value | HR (95% CI) | P value | HR (95% CI) | P value |
| Anticoagulation method | | | | | | | | |
| LMWH+ RCA vs. LMWH | 0.16 [0.04–0.67] | <0.05 | 0.13 [0.03–0.60] | <0.01 | 0.16 [0.03–0.74] | <0.05 | 0.16 [0.03–0.81] | <0.05 |
| LMWH+ RCA vs. RCA | 0.57 [0.09–3.63] | 0.55 | 0.25 [0.03–1.86] | 0.18 | 0.35 [0.04–2.88] | 0.33 | 0.21 [0.02–2.06] | 0.18 |
| Age, years | 0.99 [0.97–1.00] | 0.07 | 0.98 [0.97–1.00] | 0.07 | 1.00 [0.98–1.02] | 0.97 | 1.00 [0.98–1.03] | 0.78 |
| Male | 0.34 [0.16–0.71] | <0.01 | 0.30 [0.12–0.71] | <0.01 | 0.39 [0.12–1.21] | 0.10 | 0.33 [0.09–1.18] | 0.09 |
| BMI, kg/m$^2$ | 0.99 [0.92–1.07] | 0.85 | 1.01 [0.93–1.10] | 0.83 | 1.00 [0.91–1.10] | 0.95 | 0.99 [0.89–1.09] | 0.78 |
| ALB, g/L | 0.95 [0.90–1.00] | <0.05 | | | 1.03 [0.95–1.11] | 0.48 | 1.03 [0.95–1.11] | 0.47 |
| SCr, μmol/L | 1.00 [1.00–1.00] | 0.30 | | | 1.00 [1.00–1.01] | 0.16 | 1.00 [1.00–1.01] | 0.07 |
| eGFR, ml/min | 1.02 [1.01–1.03] | <0.01 | | | 1.02 [1.01–1.05] | <0.05 | 1.03 [1.01–1.05] | <0.01 |
| BUN, μmol/L | 0.96 [0.93–1.00] | <0.05 | | | 0.96 [0.92–1.02] | 0.17 | 0.97 [0.92–1.02] | 0.24 |
| HGB, g/L | 1.02 [1.01–1.03] | <0.01 | | | 1.01 [0.99–1.03] | 0.24 | 1.01 [1.00–1.03] | 0.17 |
| PLT, ×10$^9$/L | 1.00 [1.00–1.01] | <0.01 | | | 1.00 [1.00–1.00] | 0.97 | 1.00 [1.00–1.01] | 0.68 |
| Ca$^{2+}$, mmol/L | 0.12 [0.02–0.76] | <0.05 | | | 0.12 [0.01–2.10] | 0.15 | 0.15 [0.01–2.90] | 0.21 |
| PT, s | 0.91 [0.76–1.09] | 0.31 | | | | | 0.82 [0.61–1.11] | 0.21 |
| APTT, s | 1.02 [0.98–1.06] | 0.29 | | | | | 1.01 [0.95–1.08] | 0.73 |
| D-Dimer, μg/ml FEU | 0.96 [0.91–1.02] | 0.21 | | | | | 1.04 [0.97–1.12] | 0.28 |
| Catheter site[d] | 1.74 [0.73–4.14] | 0.21 | | | | | 1.53 [0.51–4.67] | 0.45 |
| Catheter length[e] | 0.66 [0.24–1.81] | 0.42 | | | | | 0.72 [0.18–2.90] | 0.64 |

Notes:
Abbreviations: LMWH, Low molecular weight heparin; BMI, body mass index; ALB, albumin; SCr, serum creatinine; BUN, blood urea nitrogen; HGB, hemoglobin; PLT, platelet; PT, plasma prothrombin time; APTT, activated partial thrombin time; Ca$^{2+}$, serum calcium levels.
[a] Model 1: adjusted for age, gender, baseline BMI and anticoagulation methods.
[b] Model 2: adjusted for covariates in model 1 and serum albumin, serum creatinine, eGFR (CKD-EPI), blood urea nitrogen, hemoglobin, platelet and serum calcium levels.
[c] Model 3: adjusted for covariates in model 2 and catheter site, catheter length, prothrombin time, activated partial thrombin time and D-Dimer.
[d] Catheter site was femoral vein vs. internal jugular vein.
[e] Catheter length was 20 vs. 16 cm.

complication in NS patients (*Moist, 2016*; *Lazrak et al., 2017*), often necessitates CKRT to manage volume overload (*Novak & Ellison, 2022*; *Siddall & Radhakrishnan, 2012*). Anticoagulation is a critical component of CKRT, ensuring the filter's uninterrupted function.

Heparin, including unfractionated heparin (UFH) and LMWH, is a widely utilised anticoagulant in CKRT (*Mousa, 2007*). UFH exerts its anticoagulant effects through three pathways. First, approximately one-third of the administered dose binds to antithrombin (AT), catalyzing the inactivation of factors IIa, Xa, IXa, and XIIa, which accounts for most of its anticoagulant effect. Second, the remaining two-thirds binds to heparin cofactor II, catalyzing thrombin inactivation at concentrations higher than therapeutic levels. Third, UFH binds to platelets, inhibiting their function and potentially causing heparin-induced thrombocytopenia (HIT). In contrast, LMWH primarily inactivates factor Xa through AT. Due to its smaller molecular fragments, LMWH cannot simultaneously bind to AT and thrombin, resulting in minimal inactivation of factor IIa. Compared to UFH, LMWH has a

lower incidence of HIT, a longer half-life, and is cleared primarily through the kidneys. In NS patients, the urinary loss of antithrombin III contributes to hypercoagulability and a diminished response to heparin, necessitating higher doses to achieve therapeutic anticoagulation, which in turn increases the risk of bleeding or other side effects (*Hrish et al., 2001*; *Cuker, 2016*).

Citrate's use in CKRT began in 1990 (*Mehta et al., 1990*). It only provides anticoagulation *in vitro* by chelating calcium and converts to bicarbonate *in vivo*, thus minimising bleeding risks. Substantial evidence supports its anticoagulant efficacy and safety, which can significantly extend filter lifespan and reduce complications, treatment interruptions, and costs (*Zhang & Hongying, 2012*; *Stucker et al., 2015*; *Zarbock et al., 2020*; *Oudemans-van Straaten et al., 2009*). The 2020 KDIGO guidelines advocate for RCA as the primary anticoagulant in CKRT for patients without contraindications (*Ostermann et al., 2020*). Yet, a consensus on the anticoagulation strategy for NS patients in CKRT is lacking.

The hypercoagulability associated with NS, immobilization during CKRT, and deep vein catheterization all increases the risk of thromboembolism. Literature reports a 7–50% incidence of thrombosis in NS patients, with renal vein thrombosis rates as high as 50%, and deep vein thrombosis rates between 2% and 19% (*Morgera et al., 2009*; *Lionaki et al., 2012*; *Barbour et al., 2012*). RCA does not provide *in vivo* anticoagulation. Furthermore, studies have shown that subcutaneous administration of UFH or LMWH does not affect extracorporeal circulation (*Beijering et al., 1997*). Our clinical observations suggest that hypoalbuminemia-induced hemoconcentration, which increases blood viscosity, may impair the anticoagulative efficacy of RCA. Combining RCA with LMWH may be a viable option for these patients (*Roberts et al., 2020*).

RCA and heparin anticoagulation has been explored in COVID-19 patients, prone to thromboembolism and CKRT circuit failure due to early clotting (*Chua et al., 2020*; *Valle et al., 2021*; *Shafiee et al., 2021*), which has shown superiority over either agent alone in terms of filter lifespan and clotting incidence (*Shankaranarayanan et al., 2020*; *Valle et al., 2021*; *Volbeda et al., 2020*).

Our study revealed that RCA + LMWH therapy significantly increased the median filter lifespan (22 h) and had the lowest incidence of filter clotting (5.7%) compared to LMWH (16 h, 28.1%) and RCA alone (15 h, 9.7%). This synergistic effect involves multiple factors. First, LMWH provides systemic anticoagulation by inactivating coagulation factors, primarily Xa and IIa. Second, RCA enhances extracorporeal anticoagulation by chelating calcium, maintaining low post-filter calcium levels, and disrupting the coagulation cascade *in vitro*. Finally, the combination therapy reduces filter clotting and maintains catheter patency. Catheter patency is crucial, as low blood flow or frequent interruptions in CKRT circulation caused by catheter issues significantly contribute to filter clotting. These findings also align with those of *Volbeda et al. (2020)* study. However, the median filter lifespan in COVID-19 patient studies, ranging from 72 to 81.9 h with RCA plus heparin anticoagulation (*Zarbock et al., 2020*; *Jubelirer, 1985*; *Wen et al., 2020*), far exceeds that in our study. Several factors may account for this discrepancy: the distinct pathophysiological mechanisms of hypercoagulability in NS *vs.* COVID-19 patients; the non-critical nature of our patient cohort, which did not necessitate CKRT beyond 24 h; and our protocol of

replacing filters after 24 h regardless of clotting status. Additionally, variations in heparin or LMWH administration routes, dosages, the small study population, and patient conditions may contribute to these differences. Consequently, further researches are warranted to elucidate the optimal anticoagulation effect of RCA + heparin therapy.

In the present study, we delved into the safety profile of RCA combined with LMWH in patients with NS. We observed that the administration of LMWH in conjunction with RCA resulted in a lower dosage requirement for CKRT without compromising the efficacy of CKRT or ultrafiltration rates. Notably, patients receiving the combined RCA + LMWH therapy exhibited a reduced incidence of overall adverse events and catheter-related complications, while maintaining a comparable rate of dialysis-related adverse events.

Thrombosis is a recognised complication in NS due to the disease's inherent hypercoagulability. The incidence of thrombosis was significantly lower in the RCA + LMWH group when compared to that in LMWH group. This finding is in stark contrast to patients who received RCA monotherapy, who experienced the highest incidence of thrombosis, aligning with prior research (*Wu et al., 2015*). This suggests that only combined anticoagulation will significantly mitigates the risk of thrombotic complications *in vivo* and in the circuit. RCA provides anticoagulation effects within the extracorporeal circuit without impacting systemic coagulation. The superiority of combination therapy over LMWH in reducing thrombosis lies in RCA's ability to decrease catheter clotting, a key contributor to thrombosis in CKRT patients.

Despite no significant differences in bleeding complications between different anticoagulation methods in previous researches (*Wu et al., 2015*; *Valle et al., 2021*), our study observed no bleeding events in the RCA + LMWH group, potentially attributable to the lower LMWH dosage. Due to the limited sample size, further researches are required to document the bleeding risk.

Multivariate cox proportional hazards analysis identified only lower eGFR as a protective factor against clotting. Previous studies have noted coagulopathy abnormalities in chronic kidney disease patients (*Jubelirer, 1985*), with delayed clot formation and platelet dysfunction potentially reducing clotting risk in those with kidney impairment (*Nunns et al., 2017*). Moreover, impaired kidney function prolongs LMWH metabolism, extending its anticoagulant effect (*Lim et al., 2006*; *Becker et al., 2002*; *Lim, Cook & Crowther, 2004*). These findings align with our observations that filter clotting incidence decreases in patients with kidney dysfunction, suggesting that RCA + LMWH anticoagulation is more efficacious in NS patients with normal kidney function.

This study's strength lies in its pioneering retrospective examination of RCA + LMWH anticoagulation efficacy and safety in the NS population undergoing CKRT. Our results advocate for the RCA + LMWH approach in NS patients with normal kidney function requiring extended CKRT, as it reduces filter clotting incidence and thrombotic and bleeding events.

Nonetheless, our study is not without limitations. Its single-center, retrospective cohort design and the enrollment of only 122 patients introduce potential selection bias. Moreover, the choice of anticoagulation therapy, being subject to individual physician preference, adds an element of unavoidable variability. Furthermore, the smaller sample

sizes of groups B and C may mask true intergroup differences and limit further risk factor analysis. Lack of standardized monitoring and anticoagulants dosage adjustment protocol will may compromise the reliability and consistency of the study results. Additionally, the absence of ionized calcium and other arterial blood gas (ABG) parameters precludes comprehensive adverse event documentation and calculation of cumulative dosage of citrate. Last but not least, the lack of medical imaging confirmation for thrombosis may lead to inaccuracies in incidence estimation.

## CONCLUSIONS

In conclusion, the combination of LMWH and RCA anticoagulation in CKRT for NS patients effectively extends filter lifespan, reduces clotting rates, and lowers the incidence of thromboembolism without increasing the risk of bleeding events. RCA combined with LMWH emerges as a preferable option for NS patients undergoing CKRT. Future research should aim to determine the optimal dosages of RCA and LMWH or explore alternative anticoagulation combinations.

## ACKNOWLEDGEMENTS

We extend our sincere gratitude to Guohua Zhang, Jun Liu, Jianmin Chen, Huiqing Tao and the CKRT nursing team for their invaluable contributions.

### Funding

The authors have received funding from the Science and Technology Projects in Guangzhou (grant number SL2022A04J01829 (2023A04J2307) to Diankun Liu), Nature and Science Foundation of China (grant number 82370744 to Jun Ai), and the Outstanding Youths Development Scheme of Nanfang Hospital, Southern Medical University (grant number 2017J013 to Jun Ai). The funders had no role in study design, data collection and analysis, decision to publish, or preparation of the manuscript.

### Grant Disclosures

The following grant information was disclosed by the authors:
Science and Technology Projects in Guangzhou: SL2022A04J01829 (2023A04J2307).
Nature and Science Foundation of China: 82370744.
Outstanding Youths Development Scheme of Nanfang Hospital, Southern Medical University: 2017J013.

### Competing Interests

The authors declare that they have no competing interests.

### Author Contributions

- Di Wang conceived and designed the experiments, performed the experiments, analyzed the data, prepared figures and/or tables, authored or reviewed drafts of the article, and approved the final draft.

- Mengqin Yang conceived and designed the experiments, prepared figures and/or tables, and approved the final draft.
- Siyuan Li conceived and designed the experiments, performed the experiments, analyzed the data, prepared figures and/or tables, and approved the final draft.
- Can Tang conceived and designed the experiments, analyzed the data, prepared figures and/or tables, and approved the final draft.
- Jun Ai conceived and designed the experiments, authored or reviewed drafts of the article, and approved the final draft.
- Diankun Liu conceived and designed the experiments, analyzed the data, prepared figures and/or tables, authored or reviewed drafts of the article, and approved the final draft.

## Human Ethics

The following information was supplied relating to ethical approvals (*i.e.*, approving body and any reference numbers):

The study received approval and ethical exemption from obtaining informed consent from patients due to the retrospective nature of the study from the medical ethics committee of Nanfang Hospital, Southern Medical University. The approval number is NFEC-2019-213.

## Data Availability

Data is available at Figshare:

Di Wang (2023). Data.xlsx. figshare. Dataset. https://doi.org/10.6084/m9.figshare.24599958.v1

## Supplemental Information

Supplemental information for this article can be found online at http://dx.doi.org/10.7717/peerj.18919#supplemental-information.

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
