# Peer review of "Efficacy and safety of low-molecular-weight-heparin plus citrate in nephrotic syndrome during continuous kidney replacement therapy: retrospective study"

_PeerJ, doi:10.7717/peerj.18919_

## Round 0.1 · original submission · Major Revisions

Inclusion and exclusion criterion are not clear or not well rationalized. Please address comments made by all three reviewers and provide your response in a point wise manner. Reviewer 3 has an annotated manuscript to review for authors. Manuscript needs to be thoroughly revised and needs a major revision.

·

Basic reporting

no comment

Experimental design

1. Age Group Classification: It is uncommon to classify individuals over the age of 14 as adults. Please provide a justification for using 14 years as the age cutoff in your study. Without a clear rationale, this division might appear arbitrary and could be questioned.

2. Exclusion of Unfractionated Heparin: Although you have chosen to use low-molecular-weight heparin, the exclusion of unfractionated heparin, which is modifiable and cost-effective, and can be monitored via ACT and APTT, warrants an explanation.

3. Exclusion of the No-Anticoagulant Group: Including a no-anticoagulant group could elucidate the effects of each anticoagulant more clearly, especially when assessing the combination of RCA and LMWH. Please explain the rationale for their exclusion.

4. Selection Bias: More detailed information on how anticoagulants were selected is necessary. A clear explanation of the criteria for choosing the three anticoagulant patterns would help validate the study’s findings.

5. Prophylactic Anticoagulant Therapy: If prophylactic anticoagulant therapy is a protocol at your institution for patients with nephrotic syndrome and a hypercoagulable state, please indicate this. Such information could provide important context for your study’s protocol.

6. LMWH Dosage Specification: Please specify the LMWH dosage in standardized units. Doses stated in mL can be misleading due to varying drug concentrations. Additionally, could the administered dose of LMWH have been insufficient? A comparison with standard doses used in CRRT settings would be informative.

7. Filtration Technique Standardization: Clarify whether the blood filtration technique was post-dilution or pre-dilution, and whether this method was standardized across all patients. If post-dilution, discuss the implications of increased filtration volume on circuit clotting risk.

8. Statistical Analysis Methodology: Describe how explanatory variables were chosen for inclusion in the multivariate analysis. If multiple models are used, please include indices such as AIC and R^2 to indicate model fit.

Validity of the findings

9. Dosage Differences in Combination Therapy: Why were the doses of LMWH and citrate lower in the group receiving a combination compared to other groups? Is there a protocol variation for dosing in the combination therapy group?

10. Thrombosis and Anticoagulant Dose: Discuss why Group C, which received a lower dose of LMWH, exhibited less thrombosis compared to Group A, despite LMWH’s systemic anticoagulant effects.

11. Mechanisms of Anticoagulation: Provide a detailed explanation of the pathophysiological mechanisms underlying the anticoagulant effects of citrate and LMWH. Clearly delineate the differences between these two agents. Additionally, discuss the potential synergistic effects of their combination on circuit lifespan.

Additional comments

This manuscript evaluates the impact of anticoagulants on circuit lifespan in patients with nephrotic syndrome requiring CRRT. The combination of LMWH and RCA, theoretically considered to potentially extend circuit lifespan without increasing bleeding risks, lacks supporting data in this patient population, making this a significant contribution. However, to enhance the clarity and robustness of this report, I suggest several revisions.

·

Basic reporting

no comment

Experimental design

no comment

Validity of the findings

no comment

Additional comments

This is an interesting paper which is well written and attempts to tackle a common clinical problem which is seen in both the ICU and in patients on continuous renal replacement therapy.
There are several additional problems with the analysis. The authors should consider the following concerns:
1. Why is the combining of LMWH with RCA already being used in NS patients undergoing CRRT when the efficacy and safety are uncertain?
2. Line 78, What is the date of the laboratory data?
3. Line 116, What was the definition of lifespan of the filter for patients who stop CRRT as scheduled? Supposed a treatment gone on as planned for 18 hours, and at the end of the treatment, no clotting occurred. Was the life of this filter 18 hours or 24 hours? The lifespan of the filter was the primary outcome, but the definition and how it was recorded was not very clear.
4. Line 122, Why include catheter related infection and catheter dysfunction as adverse events? How does the choice of anticoagulation method affect catheter function or catheter related infection?
5. Line 127,thrombosis is identified by the formation of a fibrin sleeve encasing the catheter or a thrombus adhering to the vessel wall or in the catheter. How was ultrasonography used here?
6. Line 178, In Group C, RCA+ LMWH was used. The doses of LMWH were certainly lower than those given only LMWH in Group A; The dosage of RCA was certainly lower than that of RCA in Group B. This secondary outcome has no value.
7. Line 214, RCA combined with LMWH has been reported to be used as anticoagulation therapy in previous study.
8. Line 296, the results do not support the conclusion of lower incidence of bleeding event in Group C.
9. The sample sizes of Group and Group C were too small, resulting in poor reliability of the results

Reviewer 3 ·

Basic reporting

Thank you for the opportunity to review the manuscript "Efficacy and safety of low-molecular-weight-heparin plus citrate in nephrotic syndrome during continuous renal replacement therapy" by Dr Di Wang et al. l. This is a retrospective review of their experience of patients using low-molecular-weight heparin with citrate vs. those with just low-molecular-weight heparin or regional citrate anticoagulation in patients with nephrotic syndrome undergoing continuous kidney replacement therapy. The manuscript is well written, but the numbers are small, and being a retrospective review, further research is needed to optimise dosing and validate these results in broader populations. The number of patients in each group is small, so the study may not be sufficiently powered.
I also suggest that the word "renal" be replaced by "kidney", in keeping with the new nomenclature.
In the section on materials and methods, I suggest the age is stated as to whether it is a mean age or a median age.
Circuit clotting also depends on the length of CRRT, which must be factored into the calculations.
The last four tables can be either stated in words in the manuscript or merged.

Experimental design

This is a retrospective analysis, and therefore, this is not applicable.

Validity of the findings

Given the small numbers in each group, the results must be validated in larger populations of patients in a prospective study.

Additional comments

Several grammatical errors need correction.

Annotated reviews are not available for download in order to protect the identity of reviewers who chose to remain anonymous.

---

## Round 0.2 · Minor Revisions

Please address the remaining issues and respond to the reviewers' queries in a point-wise manner.

·

Basic reporting

no comment

Experimental design

1. In the section describing the choice of anticoagulants, lines 130-131 state that "Anticoagulation therapy determined by clinical physicians was tailored to individual characteristics and therapeutic objectives of the patients". However, this description is overly vague. At the very least, it should be explicitly stated that contraindicated medications were not used. Furthermore, since the physician's discretionary selection of medications can introduce selection bias, it is necessary to add a limitation regarding this aspect.

2. The description of the exclusion criteria is insufficient. It is necessary to detail which patients were excluded from the study. For example, if patients taking oral anticoagulants were excluded, please specify the reasons and conditions for this exclusion.

3. It is recommended to clearly present a flowchart detailing how study participants were selected and excluded. This will enhance the transparency of the research and make it easier for readers to understand the study design.

4. In the absence of a consensus on the optimal dosing, it is necessary to explain why the dosages differ between monotherapy and combination therapy in the study population. Additionally, it appears that your institution does not monitor the effects of anticoagulation. If monitoring is not performed, please specify the criteria used to adjust dosages. It is also essential to clarify which medication is prioritized during dosage adjustments in combination therapy. If there is a protocol in use, please present it. This information is crucial for the clinical application of your study results. If there is no protocol and dosages are determined at the discretion of the attending physician, this limitation should be appropriately documented.

5. Thank you for providing the dosage of LMWH in IU. When the dosage is presented in IU or mg, it is unnecessary to also show it in ml. This approach makes the presentation of data clearer and more concise.

Validity of the findings

6. Thank you for the reanalysis that includes prophylactic anticoagulation therapy with LMWH. Although this reanalysis is included in Table S2, it should actually be presented as the main analysis result in the text. While you mention that both results were similar, it is possible that the similarity was coincidental due to misclassification and measurement biases in the original analysis. Only the more accurately evaluated reanalysis results should be presented as the main findings. Additionally, if there are differences in the prophylactic dosages and methods of administration of LMWH, these should also be clearly documented. Why is there a need to retain the original analysis results that may contain errors?

7. The authors attribute the lower incidence of thrombosis in Group C, despite lower LMWH dosages compared to Group A, to the attenuation of LMWH's effect due to ATIII loss and the synergistic effect between RCA and LMWH. However, these explanations are insufficient. To substantiate the impact of decreased ATIII activity on LMWH effectiveness, specific data on ATIII activity in both groups are required. Additionally, it is necessary to detail whether ATIII concentrates or FFP were administered to the study population. Moreover, an examination of how the synergy between RCA and LMWH affects systemic anticoagulation is needed. Given that RCA is supposed to act locally within the dialysis circuit, a detailed explanation of how it could enhance systemic anticoagulation is also required.

8. Thank you for the explanation regarding the anticoagulant mechanisms of citrate and LMWH, but it remains insufficient. It is not enough to state that heparin's anticoagulant effect is mediated through antithrombin. Specific details are needed about which coagulation factors are inhibited by heparins. Additionally, please address the differences between unfractionated heparin and LMWH, specifically regarding their points of action, pharmacokinetics, and the presence or absence of antiplatelet effects. Furthermore, the core of this study involves how citrate's chelation of calcium and LMWH's inhibition of coagulation factors may synergistically interact. More detailed explanation on this synergy is necessary.

Additional comments

Thank you to the authors for earnestly responding to the reviewers' comments, but to further enhance the quality of the report, additional comments are necessary. More detailed explanations are required for certain analytical results and methodologies. To increase data transparency and facilitate reader comprehension, please reconsider the following points. Additionally, clearer justifications for the choice of specific research methods and the reasons behind differing results are required.

Reviewer 3 ·

Basic reporting

No comment

Experimental design

No comment

Validity of the findings

No comment

Additional comments

---

## Round 0.3 · accepted · Accept

Authors have addressed all of the reviewers' comments and manuscript is ready for publication.

Reviewer 3 ·

Basic reporting

All comments have been adequately addressed.

Experimental design

Adequate

Validity of the findings

No comment

Additional comments

Nil

Annotated reviews are not available for download in order to protect the identity of reviewers who chose to remain anonymous.